# Citric Acid and Magnolol Ameliorate *Clostridium perfringens* Challenge in Broiler Chickens

**DOI:** 10.3390/ani13040577

**Published:** 2023-02-06

**Authors:** Xieying Ding, Xin Zhong, Yunqiao Yang, Geyin Zhang, Hongbin Si

**Affiliations:** State Key Laboratory for Conservation and Utilization of Subtropical Agro-Bioresources, College of Animal Science and Technology, Guangxi University, Nanning 530004, China

**Keywords:** *Clostridium perfringens*, enteritis, citric acid, magnolol

## Abstract

**Simple Summary:**

In this study, the combined inhibitory effect of citric acid (CA) and magnolol (MA) on *C. perfringens* was first confirmed. Subsequent determination of growth curves and SEM morphological observations showed that the best synergistic effect occurred at a mass ratio of 50:3. The medicine combination not only inhibited the growth of *C. perfringens*, but also damaged its cell structure at 265 μg/mL. The effect of medicine combination is more significant at 530 μg/mL. This study then used the growth performance, inflammatory levels, antioxidant capability, and changes in intestinal microbiota to comprehensively evaluate the therapeutic effect of CA and MA on *C. perfringens* challenge in chickens. The results showed that CA and MA can inhibit inflammation via the inhibition of the TLR/MyD88/NF-κB pathway and enhance the antioxidant capability via the enhancement of the Nrf2/HO-1 pathway, thus mitigating the adverse effects of *C. perfringens* challenge. Furthermore, while the medicine combination had a relatively good effect at a dosage of 0.2%, the most effective dosage for the restoration of intestinal microbiota was 0.1%. These results indicate that the addition of CA and MA to daily feed (0.1–0.2%) for chickens can function as a new protection measure for controlling *C. perfringens* challenge in chickens.

**Abstract:**

*Clostridium perfringens* (*C. perfringens*) is a common pathogenic bacterium implicated in the enteric diseases of animals. Each year, the disease is responsible for billions of dollars of losses worldwide. The development of new phytomedicines as alternatives to antibiotics is becoming a new hotspot for treating such diseases. Citric acid (CA) and magnolol (MA) have been shown to have antibacterial, antioxidant, and growth-promoting properties. Here, the bacteriostatic effects of combinations of CA and MA against *C. perfringens* were investigated, together with their effects on yellow-hair chickens challenged with *C. perfringens*. It was found that the optimal CA:MA ratio was 50:3, with a dose of 265 μg/mL significantly inhibiting *C. perfringens* growth, and 530 μg/mL causing significant damage to the bacterial cell morphology. In animal experiments, *C. perfringens* challenge reduced the growth, damaged the intestinal structure, activated inflammatory signaling, impaired antioxidant capacity, and perturbed the intestinal flora. These effects were alleviated by combined CA–MA treatment. The CA–MA combination was found to inhibit the TLR/Myd88/NF-κB and Nrf-2/HO-1 signaling pathways. In conclusion, the results suggest the potential of combined CA–MA treatment in alleviating *C. perfringens* challenge by inhibiting the growth of *C. perfringens* and affecting the TLR/MyD88/NF-κB and Nrf-2/HO-1 signaling pathways.

## 1. Introduction

*Clostridium perfringens* (*C. perfringens*) is a common pathogenic bacterium implicated in the enteric diseases of animals [1,2]. Poultries infected with *C. perfringens* often present with decreased appetite, depression, and focal bowel necrosis, leading to serious consequences for animal welfare [3] and an estimated annual expenditure of approximately $6 billion on the prevention, diagnosis, and treatment of *C. perfringens* challenge for the poultry-based industry [4].

The addition of antibiotics to feed and water has been used for some time for growth promotion and *C. perfringens* challenge prevention in animals [5]. However, continuous and heavy use of antibiotics leads to various types of environmental pollution, particularly soil and water pollution, leading to the presence of excessive residues in foods and consequences for human health [6]. In this context, green, safe, and healthy antibiotic alternatives are becoming a hot topic to be explored.

Citric acid (CA) is an organic acid found in many plant species and is widely distributed in nature [7]. CA is also a common ingredient in human and poultry foods where it is used as a normal acidulant, flavoring agent, or preservative for poultry feed, and has been found to be effective in the prevention of pathogen challenge, immune enhancement, and growth promotion as well as having antioxidant properties [8,9].

The bark of Magnolia officinalis (hou pu) is a traditional Chinese medicine that is often used for the clinical treatment of asthma, constipation, edema, bloating, malaria, and other diseases or symptoms [10]. Magnolol (MA) is a phenolic compound and active ingredient of hou pu [11] that has been found to have anti-inflammatory, antioxidant, antibacterial, antifungal, and other properties [12]. Moreover, it has been reported that MA can offer protection against pathogens for monogastric animals [13,14].

The Bao Chi Quan Shu, an ancient collection of traditional Chinese medicinal prescriptions, provides an example of the combination treatment of intestinal diseases using wu mei (smoked plum, with CA as its main active ingredient) and hou pu (with MA as its main active ingredient), but research on the combined use of CA and MA for the treatment of *C. perfringens* challenge has not been conducted yet. Therefore, this study aimed to investigate the effects of the CA–MA combination on *C. perfringens* and its protective effect against *C. perfringens* challenge in chickens to promote the utilization of natural compounds for the treatment of *C. perfringens* challenge and specifically, the medicinal value of CA and MA.

## 2. Materials and Methods

### 2.1. C. perfringens Preparation

The species type strain of *C. perfringens* was obtained as a gift from the South China Agricultural University. The *C. perfringens* sample was thawed after cryopreservation and cultured overnight at 37 °C on a cooked meat medium (Haibo, Qingdao, China). The culture was then transferred to TSC medium (Haibo) by plate streaking and cultured anaerobically for a further 24 h, after which the single colonies (appearing as black dots) were again cultured on cooked meat media until they reached logarithmic growth before then being stored for later use.

### 2.2. CA and MA Preparation

The MA powder and CA were dissolved in DMSO and sterile water to concentrations of 50 mg/mL and 10 mg/mL, respectively. The solutions were filtered through 0.22-μm membranes and were stored at 4 °C for later use. The CA and MA samples were purchased from Yuanye Biotechnology Co. Ltd. (Shanghai, China).

### 2.3. Analysis of Effects of CA and MA on C. perfringens

The minimum inhibitory concentrations (MICs) were measured as previously described with slight modifications [15]. Serial dilutions of MA and CA were added to cooked meat medium (the MA and CA concentration in the first tube was 1.28 mg/mL and 4 mg/mL) to which the bacteria were added to a concentration of 1 × 10^5^ CFU/mL. The control group was the culture medium diluent containing DMSO and sterile water, respectively (the proportion of DMSO and sterile water in the first test tube was 20%). After 16 h of anaerobic culture at 37 °C, the MIC values were determined as the lowest medicine concentration values of the tubes in which bacterial growth was not observed. Combinations of CA and MA at different multiples of the determined MIC values were prepared to determine the combination effect and filter out the best proportion.

For the growth curves, the bacterial solution was diluted (1:1000) and then cultured overnight at 37 °C on cooked meat media. The newly prepared bacterial solution was transferred to a conical flask containing 50 mL of cooked meat medium (1 × 10^5^ CFU/mL) before the addition of the CA–MA combination at the optimal ratio, resulting in different concentrations of the combined medicines (530, 265, 132.5, and 0 μg/mL, respectively). After anaerobic static culture at 37 °C, the OD600 values of the bacterial cultures were monitored at two-hourly intervals over 24 h to draw the growth curves.

For morphological assessments of *C. perfringens*, the prepared bacterial solution was diluted (1:1000) and cultured for 6 h at 37 °C on cooked meat media, after which the bacterial concentration was adjusted to 1 × 10^9^ CFU/mL. CA–MA combinations at the optimal ratio were added to the cooked meat media to the combined medicine concentrations of 530, 265, 132.5, and 0 μg/mL. After a further 6-h incubation, the cultures were centrifuged at 3500 r/min. After the removal of the supernatant, the precipitate was rinsed with PBS. The centrifugation and washing procedure was repeated three times and the precipitates were then fixed with 2.5% glutaraldehyde for 2 h at room temperature. After washing with sterile PBS to completely remove the fixative, the samples were dehydrated in an ethanol gradient (50–100%) with subsequent replacement in 100% tert-butanol for 30 min. The samples were then dried and sputtered with gold and evaluated under scanning electron microscopy (SEM, HITACHI, SU8100) to assess the effects of the different concentrations of the medicine combination on the cell morphological structures of *C. perfringens*.

### 2.4. Investigation of the Protective Effect of CA and MA on Chickens against C. perfringens

A total of 180 1-day-old male yellow-hair chickens (with an average SD of weight, purchased from Fufeng Poultry Aquaculture Co. Ltd., Qingdao, China) were fed normally for 13 days and then randomly divided into six groups on day 14. Each group comprised three replicates and each replicate comprised 10 chickens. The groups included a control group (Control, normal feed), a model group (Model, normal feed + *C. perfringens* challenge), a group treated with a low dose of the medicine combination (Low-dose, LD, normal with 1 g/kg medicine combination + *C. perfringens*), a group treated with a medium dose of the medicine combination (Medium-dose, MD, 2 g/kg + *C. perfringens*), a group treated with a high dose of the medicine combination (High-dose, HD, 3 g/kg + *C. perfringens*), and a group treated with lincomycin as the alternative (Lincomycin, normal with 20 mg/L lincomycin solution for drinking + *C. perfringens*). The amount of the above medicine additives was per kilogram of feed. The medicine combination was a mixture of CA and MA at the optimal ratio.

Chickens were fed a corn-soybean meal base diet without antibiotics. All nutrients met the Chinese broiler rearing standards (NY/T 2004). To reduce the influence of additional variables, all chickens were housed in rooms with controllable light and atmospheric conditions. The 1-day cycle included 23 h of light and 1 h of darkness. The room temperature was set to 33 °C for the first week and then reduced by 1 °C per day until it reached 26 °C. Diets and water were available ad libitum from the beginning to the completion of the experimental process.

The design of the *C. perfringens* challenge was based on previous reports with certain modifications [16,17,18]. Briefly, *C. perfringens* was cultured for 12 h at 37 °C on media and was fed to all groups except for the Control group between days 14 and 20 at 1 mL per day by oral gavage. The Control group was fed sterile medium in the same manner.

All chickens were fasted for 12 h on days 1, 14, and 21 before their feed intake, and their body weights were measured to calculate their average daily food intake (ADFI), average daily gain in BW (ADG), and feed conversion rate as the ratio of food intake to the increase in BW (FCR) between days 1–14 and days 14–21.

### 2.5. Sampling

Blood samples were collected from two randomly chosen chickens in each replicate via jugular venipuncture on day 21. The blood was centrifuged at 4000 r/min for 5 min at 4 °C to obtain the serum.

The selected chickens were then euthanized by cervical dislocation. The sample of the jejunum was collected, the anterior 2 cm of the sample was rinsed and fixed with 4% formaldehyde for later morphological analysis, while the rest was cut into pieces, placed in a centrifuge tube, and snap-frozen in liquid nitrogen. The cecal contents were collected aseptically and snap-frozen in liquid nitrogen. The serum, the tissue samples, and the cecal contents were all stored at −80 °C before analysis.

### 2.6. Pathological Sectioning

The fixed tissues were dehydrated, embedded, and sectioned. Then, sections were deparaffinized using xylene and placed on glass slides after hydration. The slides were then stained with hematoxylin and eosin (HE) to enable the examination of morphological structures. The heights and crypt depths of three intestinal villi on each slide were measured and the ratio of the height to the crypt depth of each villus was calculated.

### 2.7. ELISA for Measuring Bioactive Proteins and Peptides

The levels of IgG, IgA, IL-1β, IL-2, TNF-α, NF-κB, Nrf2, HO-1, SOD, CAT, DAO, and hs-CRP in the sera were measured using a Chicken ELISA Kit (Boyan, Nanjing, China); the levels of IgG, IgA, IL-1β, IL-2, TNF-α, NF-κB, Nrf2, HO-1, SOD, CAT, occludin, and ZO-1 in the jejunal tissue samples were measured in the same way. All measurements were performed according to the manufacturer’s instructions.

### 2.8. Real-Time qPCR

Total RNA was extracted from jejunal tissue using a GenStar Kit (GenStar Biosolutions Co., Ltd., Beijing, China), and the concentration and purity of the general RNA were measured using an ultra-micro UV–Visible spectrophotometer. The cDNA was then reverse-transcribed with a GenStar Kit. GAPDH was used as the internal reference gene, while the 2 ^−ΔΔCT^ method was used to calculate the relative gene expression. Key signaling pathway genes related to jejunal inflammation, namely, TLR-2, TLR-4, MyD88, NF-κB, and antioxidation-related genes (Nrf-2, HO-1), were measured three times in each sample. The sequences of the primers used for RT-qPCR are shown in Table 1.

### 2.9. 16S rRNA Sequencing of Cecal Contents

DNA was extracted from the cecal content samples using DNA extraction kits, in accordance with the manufacturer’s instructions. The purity of the genomic DNA was analyzed by agarose gel electrophoresis.

The universal primers for 16S T-DNA, 338F (5′-ACTCCTACGGGGAGGAGCAG-3′), and 806R (5′-GGACTACHVGGGTWTCTAAT-3′) were used to amplify the variable-region genes of the bacteria. The amplified products were recovered and purified using the AxyPrep DNA Gel Extraction Kit (SelectScience, Waltham, MA, USA), and amplified with the QuantiFluorTST Kit (Promega, Madison, WI, USA).

The sequences were constructed using the TruSeqTM DNA Sample Prep Kits (Illumina, San Diego, CA, USA) and sequencing was conducted on the Illumina Miseq PE300 platform according to the standard protocol of Meiji Biomedical Technology Co. Ltd. (Shanghai, China).

Using a clustering threshold of 97% similarity, the operational taxonomic units (OTUs) were analyzed based using UPARSE software (v7.0.1090), and Venn diagrams were created to enable alpha diversity analysis including the calculation of the Simpson and Shannon indices as well as the comparison between the proportion of the microbial community members at the phylum and genus levels. At the same time, LEfSe analysis was used to determine the statistical differences between classes.

### 2.10. Statistical Analysis

Statistical analysis was conducted using SPSS Statistics 20 (IBM Corp., Armonk, NY, USA). One-way ANOVA was conducted using Duncan’s multiple-range test. *p* < 0.05 was considered statistically significant. 16S rRNA data analysis operations in this study were performed on the cloud platform (super computer platform of Majorbio Bio-Pharm Technology Co. Ltd. https://cloud.majorbio.com (24 October 2022)).

## 3. Results

### 3.1. Analysis of Effects of CA and MA on C. perfringens

The results showed that *C. perfringens* growth was not affected by DMSO (5%), sterile water (20%), CA (<2 mg/mL as the MIC value), and MA (<40 μg/mL as the MIC value), while a relatively good synergistic effect was found at the CA:MA ratio of 50:3 (250 μg/mL CA + 15 μg/mL MA) (Table 2 and Table 3) The medicine combination inhibited the growth of *C. perfringens* and damaged the cellular morphological structures of *C. perfringens* at 265 μg/mL. However, after 16 h, *C. perfringens* began to grow slowly and had a more significant impact on *C. perfringens* at 530 μg/mL (Figure 1 and Figure 2). However, there was little effect on the cell morphology at concentrations of 265, 132.5, and 66.25 μg/mL.

### 3.2. Growth Performance of Chickens

The average daily gain (ADG), average daily feed intake (ADFI), and feed conversion ratios [Feed conversion ratios = ADFI/ADG (FCR)] of the chickens were measured on days 2, 14, and 21. No significant differences in these indices were observed between days 1 and 14; these results are therefore not shown. However, between days 14 and 21, *C. perfringens* significantly reduced the growth performance of the chickens, while this effect was alleviated in the treatment groups. In particular, Lincomycin showed the second-best FCR result and only worse than the Control, and the ADG results in the MD and Lincomycin groups showed the greatest increases. Overall, the *C. perfringens* challenge reduced the growth performance of the chickens, and the medicine treatments alleviated these effects (Figure 3).

### 3.3. Enteropathy

Chickens in the Control group showed normal villi, while significant changes were seen in the jejunal tissue of the Model group, with clear degradation of the villi. *C. perfringens* challenge increased the damage to the jejunum, and significantly reduced the height and crypt depth of the villi as well as the ratio of the height to the crypt depth. In terms of treatment, MD and Lincomycin showed the best therapeutic effect on villus height, while improvements in crypt depth were observed in all treatment groups except for Low-dose. However, the ratios of crypt height to crypt depth did not differ significantly between the treatment groups (Figure 4 and Figure 5).

### 3.4. ELISA Results

Antibody levels were determined by the measurement of IgG and IgA; antioxidant capability was demonstrated by the measurement of Nrf2, HO-1, SOD, and CAT; the degree of inflammatory response was evaluated by the measurement of IL-1β, IL-2, TNF-α, and NF-κB; the degree of intestinal damage was assessed by the measurement of DAO, hs-CRP, tight junction proteins occludin, and ZO-1.

*C. perfringens* challenge reduced the concentrations of IgG, IgA, IL-2, Nrf2, HO-1, SOD, and CAT in the serum while increasing those of IL-1β, NF-κB, TNF-α, DAO, and hs-CRP in the serum(Figure 6). These changes were alleviated in the treatment groups, with the Medium-dose group showing the best improvement. In the jejunal tissue, *C. perfringens* challenge had similar effects, while also reducing the levels of the tight junction proteins occludin and ZO-1, with the most marked effect seen in the Medium-dose group (Figure 7).

### 3.5. RT-qPCR Results

*C. perfringens* challenge significantly increased the expression of the inflammatory signaling molecules TLR-2, TLR-4, MyD88, and NF-κB in the jejunum, while significantly reducing the expression of Nrf-2 and HO-1. All of these changes were alleviated in the treatment groups, with the Medium-dose group showing the greatest improvement and the High-dose group the least (Figure 8).

### 3.6. 16S rRNA Sequencing Analysis

The 16S rRNA sequencing results of the 36 cecal content samples generated a total of 2,693,772 valid sequences after splicing, filtering, and the removal of chimeras.

Using a threshold of 97% sequence similarity, the microflora was divided into 430 OTUs, among which 133 were shared by the six groups, while 75, 42, 67, 42, 20, and 51 OTUs were unique to the Control, Model, Low-dose, Medium-dose, High-dose, and Lincomycin groups, respectively.

As shown in Figure 9, *C. perfringens* challenge increased the Simpson index, reduced the Shannon index, and led to imbalances in the gut microbiota. Both indices were improved in the treatment groups, although the Low-dose group, interestingly, showed the best improvement.

In terms of microbial distribution at the phylum level, *C. perfringens* challenge increased the relative abundance (RA) of Firmicutes and Proteobacteria but decreased that of Actinobacteriota; compared with the Model group, all treatment groups showed increased RAs of Firmicutes and Actinobacteriota and a decreased RA of Proteobacteria.

In terms of microbial distribution at the genus level, *C. perfringens* challenge increased the RA of Lactobacillus and Escherichia–Shigella, but decreased that of Ruminococcus_torques_group, norank_f__norank_o__Clostridia_UCG014, and Blautia. Compared with the Model group, the Low-dose group showed increases in the RAs of Ruminococcus_torques_group, norank_f__norank_o__Clostridia_UCG014, and Blautia, a decreased RA of Escherichia–Shigella, and a slight decrease in the Lactobacillus RA. The Medium-dose group showed increases in Lactobacillus, Ruminococcus_torques_group, and Blautia, and decreases in Escherichia–Shigella, and norank_f__norank_o__Clostridia_UCG014, while the High-dose group showed raised levels of Lactobacillus and Blautia, and reduced numbers of Escherichia–Shigella and norank_f__norank_o__Clostridia_UCG014, with minimal changes in the abundance of Ruminococcus_torques_group. Lincomycin-treated chickens were found to have an increased abundance of Ruminococcus_torques_group and Blautia, and a reduced abundance of Lactobacillus, Escherichia–Shigella, and norank_f__norank_o__Clostridia_UCG014. However, none of these changes were significant.

## 4. Discussion

It was found that the growth of *C. perfringens* was not affected by DMSO (5%), sterile water (20%), CA (<2 mg/mL, as the MIC value), and MA (<40 μg/mL, as the MIC value), while a relatively good synergistic effect was observed when CA and MA were combined at a mass ratio of 50:3 (250 μg/mL CA + 15 μg/mL MA). The medicine combination at 265 μg/mL and above the dose will affect the growth and cell morphology of *C. perfringens*. These results demonstrate that the combined use of CA and MA is effective for inhibiting *C. perfringens* growth at reduced concentrations of both individual doses.

This study also revealed that *C. perfringens* challenge damaged the jejunal tissue of the chickens. This compromised intestinal integrity would be expected to facilitate pathogenic invasion through the damaged tissue and *C. perfringens* proliferation through alterations in the intestinal environment induced by the leakage of body fluids. This damage could lead to intestinal and systemic inflammation, accompanied by reduced antioxidant capability, thereby influencing the growth and health of the chickens. Treatment with the CA–MA combination alleviated the reduced growth performance, lowered the risks of inflammation, increased the antioxidant capability, maintained the intestinal barrier, and protected the infected chickens. These results are consistent with those of other studies that showed the alleviation of *C. perfringens* challenge by organic acids, essential oils, or plant extracts, rather than antibiotics and other medicines [18,19,20,21].

Production performance is the most direct indicator of the health of chickens. *C. perfringens* challenge can substantially reduce the production performance of chickens [22,23]. In this study, the challenge of *C. perfringens* reduced ADG and improved FCR in chickens. The combined use of CA and MA effectively alleviated the decreases in production performance. It has been found that the addition of CA to the daily feed of poultry can maintain the acidic state of the intestinal tract, reduce pathogens, improve the morphology and barrier function of the intestine, and improve the ADG [24,25]. MA participates in several biological activities including antimicrobial, anti-inflammatory, and antioxidant activities, and is effective as an additive for reducing the FCR of animal breeds [26,27,28,29]. These findings were similar to our results.

Toll-like receptors (TLRs) are important components of the body’s immune response. Bacterial challenge activates TLR-2 and TLR-4m which, in turn, activate NF-κB via the signaling molecule MyD88 [30]. This leads to increased production of inflammatory factors such as IL-1β and TNF-α, resulting in inflammation. These pro-inflammatory cytokines are often considered as biomarkers of injury or challenge, and their overproduction can have detrimental effects on the body [31,32]. The activation of the TLR/NF-κB pathway by *C. perfringens* challenge has been previously reported [33,34]. Treatment with CA and MA was found to reduce the levels of TLR-2, TLR-4, MyD88, and NF-κB as well as the production of IL-1β and TNF-α, thus reducing inflammatory damage.

When pathogenic challenge occurs, the antibody levels are an indication of the ability of the body to resist the challenge. IL-2 is produced by T cells and plays an important role in the maintenance of T cell function and B cell activation [35]. The measurement of the serum and intestinal levels of IL-2, IgG, and IgA in this study showed that *C. perfringens* challenge reduced IL-2 production, which would lead to reduced T cell function, potentially decreasing the ability of B cells to secrete antibodies, and subsequent decreased concentrations of IgG and IgA, ultimately reducing the ability to counteract the bacterial challenge. Treatment with CA and MA significantly elevated the levels of IL-2, IgG, and IgA, thus enhancing the resistance to *C. perfringens*.

Oxidative stress usually accompanies *C. perfringens* challenge [36] and results from gastrointestinal inflammation caused by bacterial challenge [37]. Since the Nrf-2/HO-1 pathway and its downstream products are the main indicators of antioxidant capability [38,39], this study examined the concentrations of the principal antioxidant enzymes of chickens (i.e., Nrf-2, HO-1, SOD, and CAT). *C. perfringens* challenge significantly reduced the levels of Nrf-2, HO-1, SOD, and CAT in both the serum and jejunal tissue of the chickens, leading to a severe reduction in the antioxidant capability. Treatment with CA and MA resulted in a significant increase in the concentrations of the antioxidant enzymes and restoration of the antioxidant capability of the chickens.

*C. perfringens* challenge can compromise intestinal integrity and thus reduce growth performance via reductions in nutrient digestion and absorption [40]. The lengths and crypt depths of the intestinal villi are the most direct parameters for the evaluation of intestinal health. Increased villus heights, increased ratios of villus height to crypt depth, and decreased crypt depths can provide a greater surface area for enhanced nutrient absorption, leading to improved growth and production [41]. Tight junction proteins are important components of the physical barrier of the intestinal mucosa. They form seals between intestinal epithelial cells to prevent the invasion of pathogens and toxins [42]. Physical and chemical barriers work together to protect gut integrity. In this study, *C. perfringens* challenge significantly reduced both the villus height and the ratio of villus height to crypt depth, while increasing the crypt depths and the concentrations of the tight junction proteins occludin and ZO-1, seriously compromising the integrity of the intestinal barrier. Treatment with CA and MA restored the health of the intestine to a certain extent.

In addition, certain bioactive substances can reflect the degree of intestinal damage. DAO is an enzyme expressed by cells of the small intestine that is released into the peripheral circulation when the intestinal villi are damaged; CRP is a marker of bacterial challenge with its level proportional to the severity of the challenge. Therefore, the levels of serum DAO and CRP can reflect the degree of intestinal mucosal damage [43,44]. In this study, *C. perfringens* challenge significantly increased the concentrations of both DAO and CRP, indicating increased intestinal cell cytoplasm and damage caused by bacteria. Treatment with CA and MA protected the intestinal cells from *C. perfringens*-induced damage.

As shown by the 16S sequencing results, *C. perfringens* challenge had a relatively large impact on the cecal microbiota, indicated by the increased Shannon and Simpson indices. Combined treatment with CA and MA resulted in the relative normalization of both indices, an indication of improved health of the microflora. The significant performance of the Low-dose treatment, however, may be attributed to the extensive inhibitory effects of additives that impeded the reproduction of certain bacteria. At the genus level, *C. perfringens* challenge increased the RA of Escherichia–Shigella, a pathogenic bacterium, which has been reported to be associated with intestinal diseases [45]. The simultaneous increase in the RA of Lactobacillus was unexpected, although similar results have been reported, albeit through an unexplained mechanism [46,47]. *C. perfringens* challenge also decreased the RA of Ruminococcus_torques_group, norank_f__norank_o__Clostridia_UCG014, and Blautia: Ruminococcus_torques_group, which may be associated with growth promotion, immune capability, and health [48]; norank_f__norank_o__Clostridia_UCG014 has been found to be positively correlated with animal growth, in addition, they are all markers of intestinal health [45]. Blautiaa is a *C. perfringens* challenge-sensitive microbe that rapidly disappears in chickens in the presence of even mild *C. perfringens* challenge [49]. The low dose of the CA and MA combination was found to be most effective for restoring the intestinal microbiota, with moderate effects at medium and high doses. Our results demonstrate that the CA and MA combination can reverse imbalances in the intestinal microbiota after *C. perfringens* challenge, thus protecting the intestinal health of poultry.

## 5. Conclusions

In this study, the combined inhibitory effect of CA and MA on *C. perfringens* was first confirmed. Subsequent determination of the growth curves and SEM morphological observations showed that the best synergistic effect occurred at a mass ratio of 50:3 (250 μg/mL CA + 15 μg/mL MA). The medicine combination not only inhibited the growth of *C. perfringens*, but also damaged its cell structure at 265 μg/mL. The medicine combination had more significant effects at 530 μg/mL. We then used the growth performance, inflammatory levels, antioxidant capability, and changes in intestinal microbiota to comprehensively evaluate the therapeutic effect of CA and MA on *C. perfringens* challenge in chickens. The results showed that CA and MA can inhibit inflammation via the inhibition of the TLR/MyD88/NF-κB pathway and enhance the antioxidant capability via the enhancement of the Nrf-2/HO-1 pathway, thus mitigating the adverse effects of *C. perfringens* challenge. Furthermore, while the medicine combination had a relatively good effect at a dosage of 0.2%, the most effective dosage for the restoration of intestinal microbiota was 0.1%. These results indicate that the addition of CA and MA to daily feed (0.1–0.2%) for chickens can function as a new protection measure for controlling *C. perfringens* challenge in chickens.

## Figures and Tables

**Figure 1 animals-13-00577-f001:**
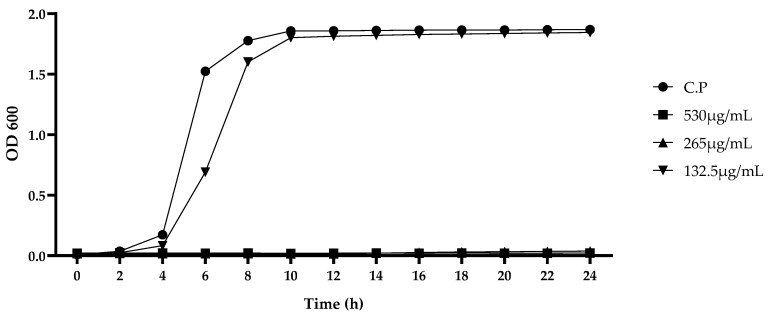
Growth curve of *C. perfringens* at different concentrations. *C. perfringens* represents the blank control.

**Figure 2 animals-13-00577-f002:**
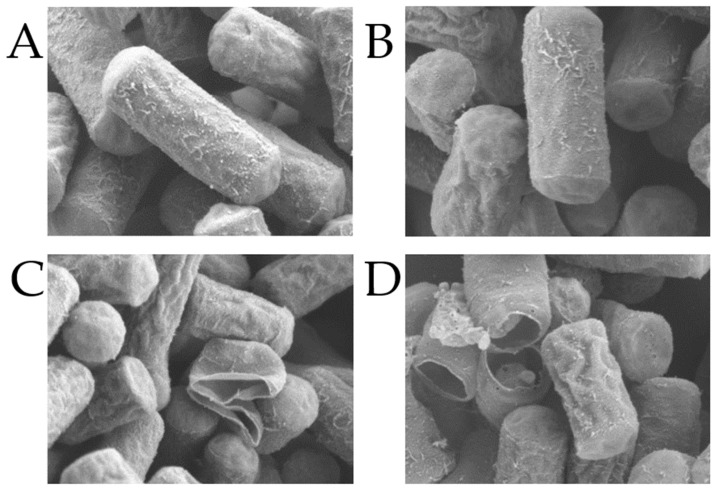
Scanning electron micrographs of *C. perfringens* at different CA–MA concentrations. (**A**) blank control; (**B**) 132.5 μg/mL; (**C**) 265 μg/mL; (**D**) 530 μg/mL.

**Figure 3 animals-13-00577-f003:**
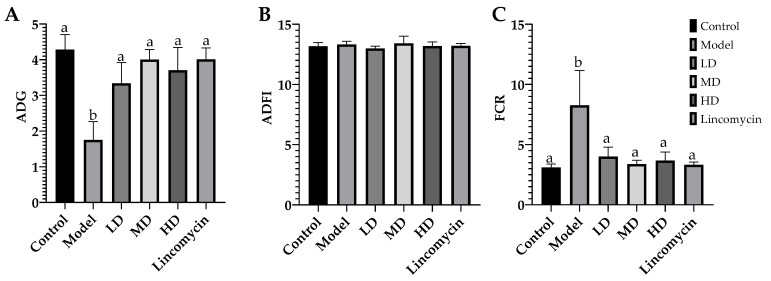
Comparison of the growth performance. (**A**) ADG; (**B**) ADFI; (**C**) FCR. Values with the same superscript letters in the same line were not significantly different (*p* > 0.05); values with different letters were significantly different (*p* < 0.05).

**Figure 4 animals-13-00577-f004:**
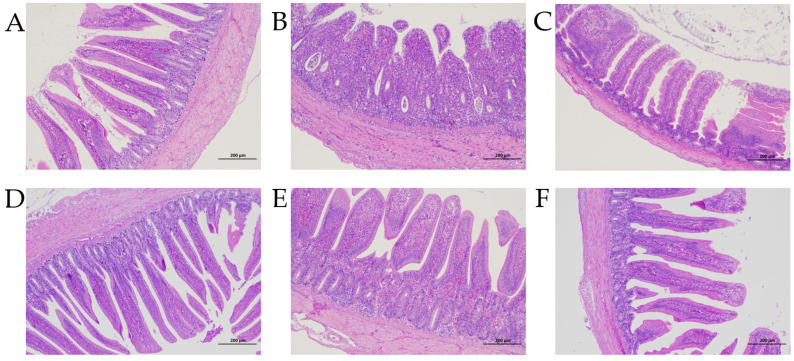
Jejunal tissue sections from each group. (**A**) Control; (**B**) Model; (**C**) Low-dose (LD); (**D**) Medium-dose (MD); (**E**) High-dose (HD); (**F**) Lincomycin.

**Figure 5 animals-13-00577-f005:**
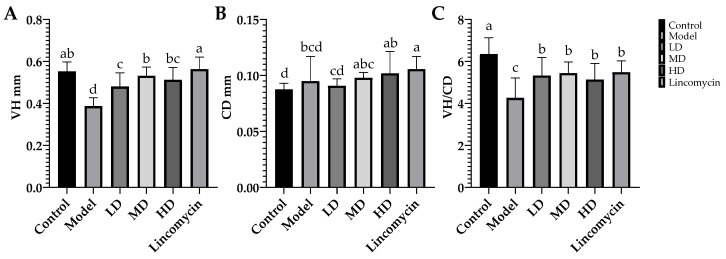
Jejunal morphological metrics. (**A**) Jejunal villus height (VH); (**B**) jejunal crypt depth (CD); (**C**) villus height/crypt depth (V/C). Values with the same superscript letters in the same line were not significantly different (*p* > 0.05); values with different letters were significantly different (*p* < 0.05).

**Figure 6 animals-13-00577-f006:**
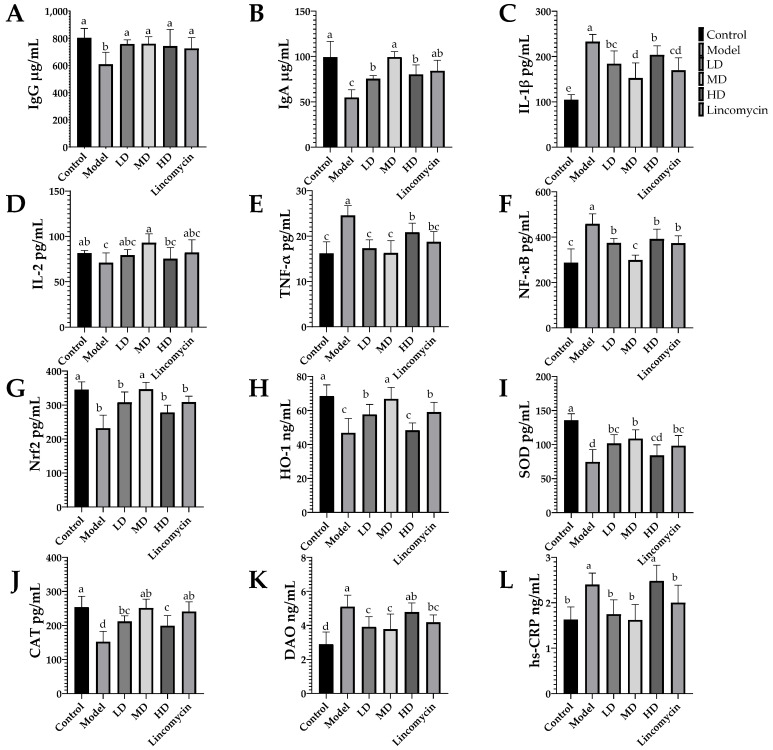
Serum concentrations of immunoglobulins, inflammatory factors, and antioxidant factors. (**A**) Immunoglobulin G (IgG); (**B**) immunoglobulin A (IgA); (**C**) interleukin 1 beta (IL-1β); (**D**) interleukin 2 (IL-2); (**E**) tumor necrosis factor alpha (TNF-α); (**F**) nuclear factor kappa B (NF-κB); (**G**) nuclear factor erythroid 2-related factor 2 (Nrf-2); (**H**) heme oxygenase 1 (HO-1); (**I**) superoxide dismutase (SOD); (**J**) catalase (CAT); (**K**) diamine oxidase activity (DAO); (**L**) C-reactive protein (CRP). Values with the same superscript letters in the same line were not significantly different (*p* > 0.05); values with different letters were significantly different (*p* < 0.05).

**Figure 7 animals-13-00577-f007:**
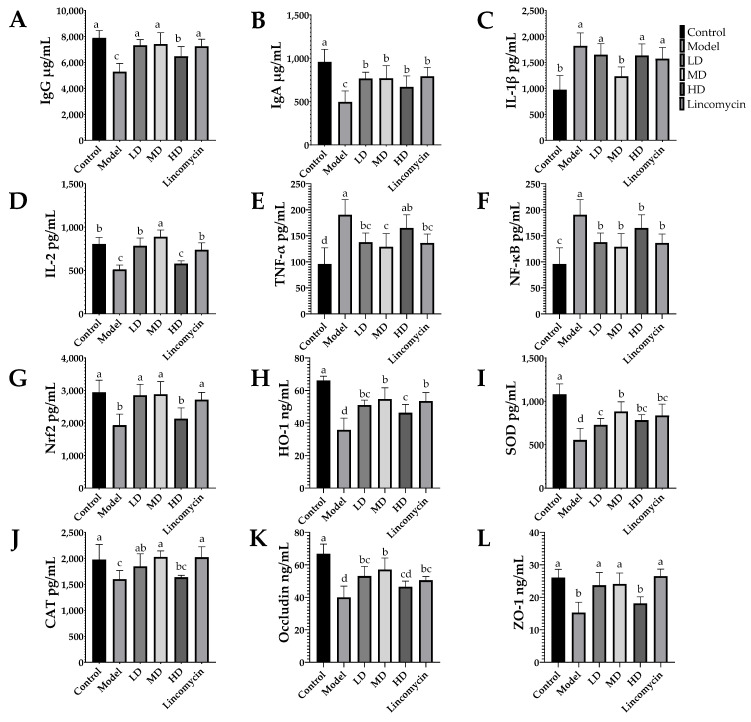
Concentrations of various immunoglobulins, inflammatory factors, and antioxidant factors in jejunal tissue. (**A**) Immunoglobulin G (IgG); (**B**) immunoglobulin A (IgA); (**C**) interleukin 1 beta (IL-1β); (**D**) interleukin 2 (IL-2); (**E**) tumor necrosis factor alpha (TNF-α); (**F**) nuclear factor kappa B (NF-κB); (**G**) nuclear factor erythroid 2-related factor 2 (Nrf-2); (**H**) heme oxygenase 1 (HO-1); (**I**) catalase (CAT); (**J**) superoxide dismutase (SOD); (**K**) occludin; (**L**) zonula occludens 1 (ZO-1). Values with the same superscript letters in the same line were not significantly different (*p* > 0.05); values with different letters were significantly different (*p* < 0.05).

**Figure 8 animals-13-00577-f008:**
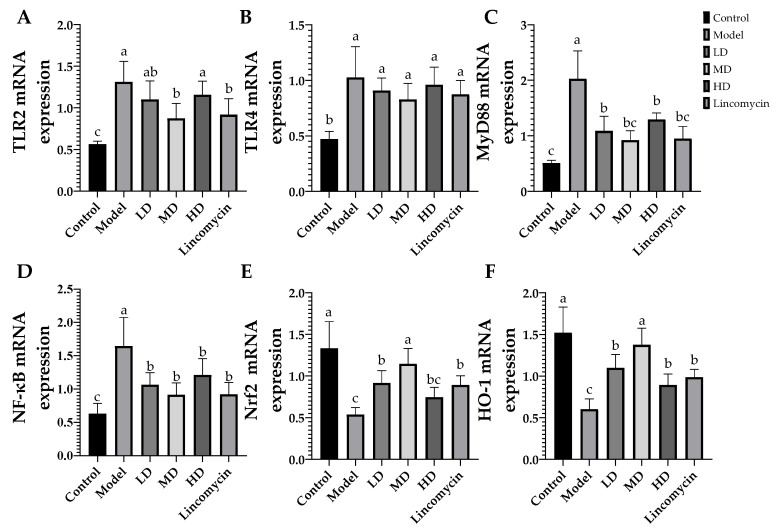
Relative mRNA expression of inflammation- and antioxidation-associated factors in the jejunal tissue. (**A**) Toll-like receptor 2 (TLR-2); (**B**) Toll-like receptor 4 (TLR-4); (**C**) myeloid differentiation primary response protein (MyD88); (**D**) nuclear factor kappa B (NF-κB); (**E**) nuclear factor erythroid 2-related factor 2 (Nrf-2); (**F**) heme oxygenase 1 (HO-1). Values with the same superscript letters in the same line were not significantly different (*p* > 0.05); values with different letters were significantly different (*p* < 0.05).

**Figure 9 animals-13-00577-f009:**
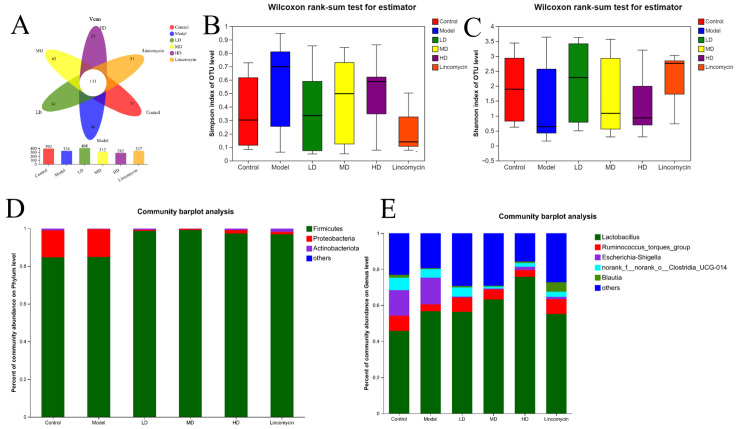
16S rRNA sequencing results. (**A**) Venn diagram; (**B**) Simpson index; (**C**) Shannon index; (**D**) differences at the phylum level; (**E**) differences at the genus level.

**Table 1 animals-13-00577-t001:** Primer sequences used for RT-qPCR.

Gene Name	Primer Sequence (5′→3′)
TLR2	F:GGGGCTCACAGGCAAAATC
	R:AGCAGGGTTCTCAGGTTCACA
TLR4	F:AGTCTGAAATTGCTGAGCTCAAAT
	R:GCGACGTTAAGCCATGGAAG
MyD88	F:GAAGTTGGGCCACGACTACCT
	R:TTGCACTTGACCGGAATCAGC
NF-κB	F:TGACCGCCAATAGCTTGTCC
	R:ACAGCTAAATGCAATGCCGTTC
Nrf-2	F:GGGCAAGGCGTGAAGTTTTT
	R:GGCTTTCTCCCGCTCTTTCT
HO-1	F:AGCTTCGCACAAGGAGTGTT
	R:GGAGAGGTGGTCAGCATGTC

TLR-2, Toll-like receptor 2; TLR-4, Toll-like receptor 4; MyD88, myeloid differentiation factor 88; NF-κB, nuclear factor kappa B; Nrf2, nuclear factor erythroid 2-related factor 2; HO-1, heme oxygenase 1.

**Table 2 animals-13-00577-t002:** MIC of CA and MA on *C. perfringens*.

Medicine	*C. perfringens*
	MIC
MA	40 μg/mL
CA	2000 μg/mL

*C. perfringens*, *Clostridium perfringens*; MA, magnolol; CA, citric acid.

**Table 3 animals-13-00577-t003:** CA and MA best proportion screening.

Different Multiples of MIC	*C. perfringens*		
	MIC of MA	MIC of CA	FICI
4 × MA − 1 × CA	20 μg/mL	250 μg/mL	0.625
3 × MA − 1 × CA	15 μg/mL	250 μg/mL	0.5
2 × MA − 1 × CA	20 μg/mL	500 μg/mL	0.75
1 × MA − 1 × CA	20 μg/mL	1000 μg/mL	1
1 × MA − 2 × CA	10 μg/mL	1000 μg/mL	0.75
1 × MA − 3 × CA	10 μg/mL	1500 μg/mL	1

*C. perfringens*, *Clostridium perfringens*; MA, magnolol; CA, citric acid. $\mathrm{FICI}=\frac{\mathrm{MIC}\;\left(A\;\mathrm{in}\;\mathrm{the}\;\mathrm{presence}\;\mathrm{of}\;B\right)}{\mathrm{MIC}\;\left(A\;\mathrm{alone}\right)}+\frac{\mathrm{MIC}\;\left(B\;\mathrm{in}\;\mathrm{the}\;\mathrm{presence}\;\mathrm{of}\;A\right)}{\mathrm{MIC}\;\left(B\;\mathrm{alone}\right)}.$

## Data Availability

The datasets generated during and/or analyzed during the present study are available from the corresponding author upon reasonable request.

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
