# Peer review of "Citric Acid and Magnolol Ameliorate Clostridium perfringens Challenge in Broiler Chickens"

_animals, 2023, doi:10.3390/ani13040577_

Round 1

Reviewer 1 Report

animals-2121290

Two natural substances as antibiotic alternatives ameliorate Clostridium perfringens challenged

This is an interesting research; however, some revision is required to improve the understanding and the quality for publication.

See below some comments:

Title: please improve the title. It is important to indicate that this study was conducted in broilers and please indicate the name of substances (Citric acid and magnolol) instead of saying just too natural substances

Please also avoid using as antibiotic alternatives

L8 – citric acid

L9 - Clostridium perfringens in italic always

(CP) don’t italicize

Please review all your document

L11 – drug? Probably you have a better word to indicate the components

Please review all document as well

L11 – delete both

L12 – I don’t like personnel language We then used

L18 and L19: %

L19 Please correct sentence “A to daily feed for chickens can function as a new alternative therapy”. Do you think this is a therapy?

L35 citric acid; magnolol

L39 - Poultry with… there are better ways to say this as symptoms

L49-59 – all mentions are in humans and animals in general. If possible, please indicate the effects in poultry or monogastric animals. It will be more useful for your introduction

The objective should be improved

Please be consistent with broilers or broiler chickens.

2.1. Analysis of effects of CA and MA on CP - Can you say citric acid and magnolol preparation?

And then Clostridium perfringens inoculum as a subtitle to be included before the last paragraph in this item

L192-194- Do you used the same statistical analysis for 16s? please indicate more details about this data analysis

3.1. Analysis of effects of CA and MA on CP, please improve this title as well

It can be i.ie Effects of CA and MA on Clostridium perfringens challenged broilers

L204 – please be consistent with C. perfringens. Actually, I prefer C. perfringens instead of CA that is not usual

Tables: you should include a footnote for the abbreviations. Same comment for figures

Figures 5-8. Please increase letter size

Figure 9. please increase quality

Author Response

Dear Reviewer,

Thanks very much for taking your time to review this manuscript. I really appreciate all your comments and suggestions! Please find my itemized responses in below and my revisions/corrections in the re-submitted files.

Please note that the number of lines referred to here is not the revision mode version. And according to the suggestions of you and other reviewers, a new Simple Summary has been prepared. I'm sorry to bother you to review this part again.

Thanks again!

Point 1: Title: please improve the title. It is important to indicate that this study was conducted in broilers and please indicate the name of substances (Citric acid and magnolol) instead of saying just too natural substances. Please also avoid using as antibiotic alternatives

Response 1: Thank you for your valuable comments. Now the title has been changed to “Citric Acid and Magnolol Ameliorate Clostridium perfringens Challenge in Chickens”, please check.

Point 2: L8 – citric acid

L9 - Clostridium perfringens in italic always(CP) don’t italicize

Please review all your document

L11 – drug? Probably you have a better word to indicate the components

Please review all document as well

L11 – delete both

L12 – I don’t like personnel language We then used

L18 and L19: %

L19 Please correct sentence “A to daily feed for chickens can function as a new alternative therapy”. Do you think this is a therapy?

L35 citric acid; magnolol

L39 - Poultry with… there are better ways to say this as symptoms

The objective should be improved

Please be consistent with broilers or broiler chickens.

Response 2: I'm sorry that I made so many mistakes. Thank you for pointing out. I have corrected them all now.

Point 3: L49-59 – all mentions are in humans and animals in general. If possible, please indicate the effects in poultry or monogastric animals. It will be more useful for your introduction

Response 3: Thank you very much for your review. The references to animals in this part are chicken and monogastric animals, and the original expression is not clear enough. The article has revised the relevant description. You can find the modified part in lines 52-62.

Point 4: 2.1. Analysis of effects of CA and MA on CP - Can you say citric acid and magnolol preparation?

 And then Clostridium perfringens inoculum as a subtitle to be included before the last paragraph in this item

Response 4: I'm sorry I didn't completely follow your suggestions. I now divide the original 2.1 part into 2.1. CP preparation, 2.2. CA and MA preparation, 2.3. Analysis of effects of CA and MA on CP. You can find the modified part in lines 73-86.

Point 5: L192-194- Do you used the same statistical analysis for 16s? please indicate more details about this data analysis

Response 5: I'm sorry I didn't explain the 16s analysis method. Thank you for your careful review. I have now added the analysis method. You can find the modified part in lines 204-206.

Point 6: 3.1. Analysis of effects of CA and MA on CP, please improve this title as well

Response 6 : Thank you for your valuable suggestions. However, since I split the 2. 1. parts and retained the title of " Analysis of effects of CA and MA on CP ". This part includes the effects of CA and MA on the inhibition concentration, growth and cell morphology of Clostridium perfringens. Therefore, I think the original title may be appropriate.

Point 7: L204 – please be consistent with C. perfringens. Actually, I prefer C. perfringens instead of CA that is not usual

Response 7 : Thank you for your proposal. I have changed all CP to C. perfringens.

Point 8: Tables: you should include a footnote for the abbreviations. Same comment for figures

Figures 5-8. Please increase letter size

Figure 9. please increase quality

Response 8 : I have added footnotes and modified the picture. Please check.

Reviewer 2 Report

The authors tackle an important infection for the poultry industry, due to the consequences for animal welfare and health, human health and also for the economy.

From One Health perspective, the research tends to explore the potential of a natural combination of compounds as an alternative to antibiotics, also envisaging environment health, therefore it has a further increased importance.

Review of

Two Natural Substances as Antibiotic Alternatives Ameliorate  Clostridium perfringens Challenged  Xieying Ding 1 , Xin Zhong 1 , Yunqiao Yang 1 , Geyin Zhang 1 and Hongbin Si 1, *

In the title: did the authors mean “challenge” instead of “Challenged”?

Line 20: replace “challenged” with “challenge” or rephrase “CP challenged in chickens” in “CP in challenged chickens”.

Line 27: replace “with CP challenged” with “challenged with CP”

Line 29: replace “CP challenged” with “CP challenge”

Line 33: replace “CP challenged” with “CP challenge”; since this mistake is repeated, the authors should consider either rephrasing the sentences of replacing “CP challenged” wherever it is the case with “CP challenge”  (line 39, 42, 44, 52, 64, 65 …etc)

Lines 44-46: could the authors reconsider the statement by the timely order in the sentence (ie, “However, continuous and heavy use of antibiotics leads to the presence of excessive residues in foods and consequences  for human health and also various types of environmental pollution, particularly soil and water pollution[6].

Line 52: what kind of “challenge”?

Lines 81-86: this part “The minimum inhibitory concentrations (MICs) were measured as previously described with slight modifications [15]. Serial dilutions of MA and CA were added to cooked meat medium (individual CA and MA compound concentrations????) to which the bacteria were added to a concentration of 1*105 CFU/m. The control group contained dilutions of DMSO only (no control with sterile water???). After 16 h of anaerobic culture at 37℃, the MIC values were determined as the lowest drug concentration values  of the tubes in which bacterial growth was not observed.” should be more clearly explained

Line 88: “the optimal ratio was found to be 50:3 according to the tests” – could the procedure be explained?

Lines 93;99: Why the concentration of 530 μg/ml of the mixed CA and MA was not tested while evaluating the growth curve, but in the morphological assessment while the rest of the concentrations match in the two procedures?

Line 103: glutaraldehyde concentration?

Lines 115-118: The dosage of the combined drugs is related to the chicken body weight or the feed (1g/kg, 2g/kg, 3g/kg)?

Lines 126-127: “Normal food and water were provided for all chickens from the beginning to the completion of the experimental process.”

What does normal mean and how was the food provided, ad libitum or following a certain ratio?

Lines 143-144: the sentence “The cecal contents were collected aseptically and snap-frozen in liquid nitrogen.” replicates

Line 148: there is no reference in line 141 to further processing of the sections preserved in formaldehyde (“The sections were deparaffinized using xylene….”)

Lines 154-158: a Chicken kit? (“The levels of IgG, IgA, IL-1β, IL-2, TNF-α, NF-κB, Nrf2, HO-1, SOD, CAT, DAO, and 154 hs-CRP in the sera were measured using a Chicken ELISA kit (Boyan, Nanjing); the levels 155 of IgG, IgA, IL-1β, IL-2, TNF-α, NF-κB, Nrf2, HO-1, SOD, CAT, occludin, and ZO-1 in 156 jejunal tissue samples was measured in the same way. All measurements were performed 157 according to the manufacturer’s instructions.”) Maybe a more detailed description could be useful

Lines 197-198: these concentrations are not mentioned in the Materials and Methods (“The results showed that CP growth was not affected by DMSO (5%), CA (<2mg/mL as the MIC value), and MA (<40µg/mL as the MIC value…”)

Table 2: Drug not Durg

Table 3. Where is FICI explained?

Fig 1 page 6: it is difficult to sense that there are two lines (530μg/ml and 265μg/ml) overlapping on Ox axis at a normal size of the graph, should be better explained in the text

Fig 2. In C the bacteria look pretty damaged

Line 213: “The ADG, ADFI, and FCR” – the abbreviations should be explained in the text before being placed in the legend of the figure

Fig. 5: One can presume the sequence is from the left to the right since there is no indication below each graph (lines 240-242)

(“Figure 5. Jejunal morphological metrics. A, jejunal villus height (VH); B, jejunal crypt depth (CD); C, villus height/crypt depth (V/C). Values with the same superscript letters in the same line are not  significantly different ( P> 0.05); values with different letters are significantly different ( P< 0.05).”)

Lines 321-324: “The drug combination inhibited CP growth and doses of 265 μg/ml and above resulted in significant morphological changes in the intestines of the chickens. These results demonstrated that the combined use of CA and MA is effective for inhibiting CP growth at reduced concentrations of both individual doses.” – no such dosage is indicated in the in vivo administration protocol

Lines 336-344: The productive performance, although there are plenty of own results, is discussed only in general terms

Lines 394-412: Although the changes in the bacterial balance is broadly discussed, there is no reference of the relevance of the identified genera and their changes in various groups on the health of chickens

Lines 417-418: Presumably, in discussion is the morphological damage of the intestinal epithelium, not of the bacteria (“The drug combination both inhibited CP growth and reversed intestinal morphological damage at 265 μg/ml”); again, in the administration protocol of the combined CA-MA there is no such dosage

Conclusion, lines 425-427: “These results indicate that the addition of CA and MA (could a real conclusion on the optimal dosage from all points of view be placed here????) to daily feed for chickens can function as a new alternative  therapy for controlling CP challenged in chicken”

Author Response

Dear Reviewer,

Thanks very much for taking your time to review this manuscript. I really appreciate all your comments and suggestions! Please find my itemized responses in below and my revisions/corrections in the re-submitted files.

Please note that the number of lines referred to here is not the revision mode version. And according to the suggestions of you and other reviewers, a new Simple Summary has been prepared.. I'm sorry to bother you to review this part again.

Thanks again!

Point 1: In the title: did the authors mean “challenge” instead of “Challenged”?

Response 1: Thank you for pointing out my writing mistakes. I have now revised the title according to your comments and those of other reviewers. Please check.

Point 2: Line 20: replace “challenged” with “challenge” or rephrase “CP challenged in chickens” in “CP in challenged chickens”.

Line 27: replace “with CP challenged” with “challenged with CP”

Line 29: replace “CP challenged” with “CP challenge”

Line 33: replace “CP challenged” with “CP challenge”; since this mistake is repeated, the authors should consider either rephrasing the sentences of replacing “CP challenged” wherever it is the case with “CP challenge”  (line 39, 42, 44, 52, 64, 65 …etc)

Lines 143-144: the sentence “The cecal contents were collected aseptically and snap-frozen in liquid nitrogen.” Replicates

Response 2: Thank you again for pointing out my writing mistakes. I have corrected them all now. Please check.

Point 3: Lines 44-46: could the authors reconsider the statement by the timely order in the sentence (ie, “However, continuous and heavy use of antibiotics leads to the presence of excessive residues in foods and consequences  for human health and also various types of environmental pollution, particularly soil and water pollution[6].”

Response 3: Thank you for pointing out the description of the sentence. Thank the Animals editorial department for correcting the description for me. You can find the modified part in lines 47-50.

Point 4: Line 52: what kind of “challenge”?

Response 4: Please forgive my inaccurate description. It is now explained in the article. It is “pathogen challenged”. You can find the modified part in lines 55.

Point 5: Lines 81-86: this part “The minimum inhibitory concentrations (MICs) were measured as previously described with slight modifications [15]. Serial dilutions of MA and CA were added to cooked meat medium (individual CA and MA compound concentrations????) to which the bacteria were added to a concentration of 1*105 CFU/m. The control group contained dilutions of DMSO only (no control with sterile water???). After 16 h of anaerobic culture at 37℃, the MIC values were determined as the lowest drug concentration values  of the tubes in which bacterial growth was not observed.” should be more clearly explained

Lines 197-198: these concentrations are not mentioned in the Materials and Methods (“The results showed that CP growth was not affected by DMSO (5%), CA (<2mg/mL as the MIC value), and MA (<40µg/mL as the MIC value…”)

Response 5: Thank you for pointing out that my language is not precise. I have now corrected the experimental description. You can find the modified part in lines 87-96 and 209-210.

Point 6: Line 88: “the optimal ratio was found to be 50:3 according to the tests” – could the procedure be explained?

Response 6: Thank you for your careful review. The ratio of 50:3 is the best ratio determined according to FICI results. This description is used for convenience in writing. The description has now been changed to make it more consistent with the article structure. You can find the modified part in lines 94-96.

Point 7: Lines 93;99: Why the concentration of 530 μg/ml of the mixed CA and MA was not tested while evaluating the growth curve, but in the morphological assessment while the rest of the concentrations match in the two procedures?

Response 7: I am very sorry that this is a mistake in my writing. The correct concentrations are 530 μg/ml, 265 μg/ml and 132.5 μg/ml. It has now been corrected. You can find the modified part in lines 101, 107-108.

Point 8: Line 103: glutaraldehyde concentration?

Response 8: Thank you again for your careful review.The glutaraldehyde concentration is 2.5%. You can find the modified part in lines 111.

Point 9: Lines 115-118: The dosage of the combined drugs is related to the chicken body weight or the feed (1g/kg, 2g/kg, 3g/kg)?

Response 9: Thank you for your review. The dosage of the combined drugs is related to the feed. I have now added a description. You can find the modified part in lines 128-129.

Point 10: Lines 126-127: “Normal food and water were provided for all chickens from the beginning to the completion of the experimental process.”

What does normal mean and how was the food provided, ad libitum or following a certain ratio?

Response 10: Thank you for your review. Diets and water were available ad libitum from the beginning to the completion of the experimental process. I have now added a description in the article. You can find the modified part in lines 136-137.

Point 11: Line 148: there is no reference in line 141 to further processing of the sections preserved in formaldehyde (“The sections were deparaffinized using xylene….”)

Response 11: Thank you for your hard work. I have now completed the experimental steps. You can find the modified part in lines 157.

Point 12: Lines 154-158: a Chicken kit? (“The levels of IgG, IgA, IL-1β, IL-2, TNF-α, NF-κB, Nrf2, HO-1, SOD, CAT, DAO, and 154 hs-CRP in the sera were measured using a Chicken ELISA kit (Boyan, Nanjing); the levels 155 of IgG, IgA, IL-1β, IL-2, TNF-α, NF-κB, Nrf2, HO-1, SOD, CAT, occludin, and ZO-1 in 156 jejunal tissue samples was measured in the same way. All measurements were performed 157 according to the manufacturer’s instructions.”) Maybe a more detailed description could be useful

Response 12: Thank you again for your careful review. However, I can't fully understand your suggestion. The kit used for Elisa detection in this experiment is a special kit for chicken. I hope my answer can explain this question.

Point 13: Table 2: Drug not Durg

Response 13: Thank you for pointing out my spelling mistakes. I have changed “drug” into “medicine” according to the opinions of other reviewers.

Point 14: Table 3. Where is FICI explained?

Response 14: I'm sorry I didn't explain FICI's calculation method. It has been added in the article, please refer to. You can find the modified part in lines 220-221.

Point 15: Fig 1 page 6: it is difficult to sense that there are two lines (530μg/ml and 265μg/ml) overlapping on Ox axis at a normal size of the graph, should be better explained in the text

Fig 2. In C the bacteria look pretty damaged

Response 15: I'm sorry that my description at the beginning is not accurate enough. The description has now been modified again: The medicine combination inhibited the growth of C. perfringens and damaged to cel-lular mor-phological structures of C. perfringens at 265 μg/ml. But after 16 hours, C. perfringens began to grow slowly. And it has more significant impact on C. perfringens at 530μg/ml. You can find the modified part in lines 212-215.

Point 16: Line 213: “The ADG, ADFI, and FCR” – the abbreviations should be explained in the text before being placed in the legend of the figure

Response 16: Thank you for pointing out the problem of the article. Now the description of ADG, ADFI and FCR has been added in the body. You can find the modified part in lines 230-231.

Point 17: Fig. 5: One can presume the sequence is from the left to the right since there is no indication below each graph (lines 240-242)

(“Figure 5. Jejunal morphological metrics. A, jejunal villus height (VH); B, jejunal crypt depth (CD); C, villus height/crypt depth (V/C). Values with the same superscript letters in the same line are not  significantly different ( P> 0.05); values with different letters are significantly different ( P< 0.05).”)

Response 17: Thank you again for pointing out the shortcomings. I have now modified the picture and added a number. Please check.

Point 18: Lines 321-324: “The drug combination inhibited CP growth and doses of 265 μg/ml and above resulted in significant morphological changes in the intestines of the chickens. These results demonstrated that the combined use of CA and MA is effective for inhibiting CP growth at reduced concentrations of both individual doses.” – no such dosage is indicated in the in vivo administration protocol

Lines 417-418: Presumably, in discussion is the morphological damage of the intestinal epithelium, not of the bacteria (“The drug combination both inhibited CP growth and reversed intestinal morphological damage at 265 μg/ml”); again, in the administration protocol of the combined CA-MA there is no such dosage

Response 18: This is a clerical error. I'm very sorry for such mistakes. The description of the problem has been changed to: The medicine combination at 265μg/ml and above the dose will affect the growth and cell morphology of C. perfringens. You can find the modified part in lines 343-344 and 442-443.

Point 19: Lines 336-344: The productive performance, although there are plenty of own results, is discussed only in general terms

Response 19: Thank you very much for your suggestion. I have added the corresponding term. Please check. You can find the modified part in lines 358-367.

Point 20: Lines 394-412: Although the changes in the bacterial balance is broadly discussed, there is no reference of the relevance of the identified genera and their changes in various groups on the health of chickens

Response 20: Your suggestion is very valuable. I apologize for the incompleteness of the statement at that time. Now relevant descriptions have been added in the article. Please review. You can find the modified part in lines 418-437.

Point 21: Conclusion, lines 425-427: “These results indicate that the addition of CA and MA (could a real conclusion on the optimal dosage from all points of view be placed here????) to daily feed for chickens can function as a new alternative therapy for controlling CP challenged in chicken”

Response 21: Thank you very much for your suggestion. But I can't give an accurate addition measurement. I can only make sure that the addition amount is between 0.1% and 0.2%. You can find the modified part in lines 452.

Round 2

Reviewer 1 Report

Why you decide to use medicine combination?

Did you use the same type of letter in all figures and text?

Title: Citric Acid and Magnolol Ameliorate Clostridium perfringens Challenge in Broiler Chickens

Author Response

Dear reviewer

Thank you for reviewing our manuscript again in your busy schedule.

We revised the manuscript according to these comments and suggestions. In general, we have tried our best to revise our manuscript and provide the point-by-point responses.

By the way, I accepted all the changes made last time. All modifications are the same as the last time. We think it is convenient for you to review again.

Thanks again!

Point 1: Why you decide to use medicine combination?

Response 1: Your question is very professional. Thank you for your careful review.

In ancient Chinese medicine books, it is recorded that the combination of wu mei (smoked plum) and hou pu (magnolia officinalis) can treat intestinal diseases.

Therefore, this study started from this point and explored the main active ingredients of two herbal medicines to alleviate the infection of Clostridium perfringens.

In this study, we first explored the antibacterial part, and found that citric acid and magnolol had the effect of inhibiting Clostridium perfringens, and the antibacterial concentration of magnolol was lower.

Then we explored the best ratio of the two medicines, hoping to reduce the amount of medicines added and reduce the feeding cost.

Fortunately, we found the ratio. In the experiment of chicken infection, we found that the medicine combination had a good effect of alleviating the challenge, which was comparable to that of lincomycin.

In conclusion, the exploration process of this study ultimately determines the use of medicine combination.

Point 2: Did you use the same type of letter in all figures and text?

Response 2: I'm sorry I didn't pay attention to the unity of the two parts. Thank you very much for pointing out this flaw. I have unified type now, please check.

Point 3: Title: Citric Acid and Magnolol Ameliorate Clostridium perfringens Challenge in Broiler Chickens

Response 3: Thank you very much for your suggestions on the topic. I have changed the title according to your suggestion.
